# Kernel-Based Regularized EEGNet Using Centered Alignment and Gaussian Connectivity for Motor Imagery Discrimination

Mateo Tobón-Henao *, Andrés Marino Álvarez-Meza and Cesar German Castellanos-Dominguez

Signal Processing and Recognition Group, Universidad Nacional de Colombia, Manizales 170003, Colombia; amalvarezme@unal.edu.co (A.M.Á.-M.); cgcastellanosd@unal.edu.co (C.G.C.-D.)
* Correspondence: mtobonh@unal.edu.co

**Abstract:** Brain–computer interfaces (BCIs) from electroencephalography (EEG) provide a practical approach to support human–technology interaction. In particular, motor imagery (MI) is a widely used BCI paradigm that guides the mental trial of motor tasks without physical movement. Here, we present a deep learning methodology, named kernel-based regularized EEGNet (KREEGNet), leveled on centered kernel alignment and Gaussian functional connectivity, explicitly designed for EEG-based MI classification. The approach proactively tackles the challenge of intrasubject variability brought on by noisy EEG records and the lack of spatial interpretability within end-to-end frameworks applied for MI classification. KREEGNet is a refinement of the widely accepted EEGNet architecture, featuring an additional kernel-based layer for regularized Gaussian functional connectivity estimation based on CKA. The superiority of KREEGNet is evidenced by our experimental results from binary and multiclass MI classification databases, outperforming the baseline EEGNet and other state-of-the-art methods. Further exploration of our model's interpretability is conducted at individual and group levels, utilizing classification performance measures and pruned functional connectivities. Our approach is a suitable alternative for interpretable end-to-end EEG-BCI based on deep learning.

**Keywords:** brain–computer interfaces; electroencephalography; motor imagery; regularizer; centered kernel alignment; functional connectivity; deep learning

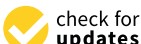



## 1. Introduction

Brain–computer interface (BCI) has emerged as a cutting-edge technology that directly connects the human brain and external devices, bridging the ultimate frontier between humans and computers [1]. This breakthrough technology has enabled people with neuromotor disorders, nervous system injuries, or limb amputations to control machines using their brains, as no peripheral nerves or muscles are involved in the process [2]. Motor imagery (MI) is one of the essential branches of BCIs' control paradigms, which allows users to control robots or external machines merely by imagining movement without the intervention of peripheral nerves [3]. Regarding this, BCI technology has significant potential in motor function rehabilitation [4], assistance [5], and other areas, sparking extensive discussions on MI-based approaches [6].

Acquiring signals related to brain activity is a critical aspect of MI-based BCIs, and multichannel time series signals, such as EEG, are commonly preferred due to their high time resolution, cost-effectiveness, and user-friendliness compared with other neuroimaging methods [7]. Moreover, using multichannel time series signals in MI tasks is essential as it captures the activation of multiple brain regions, enabling a comprehensive understanding of complex neural activity [8]. These signals facilitate exploring functional connectivity (FC) and coordinated patterns between brain regions during MI while reducing noise and artifacts through redundancy and robust signal processing techniques [9]. Nonetheless, the insufficient functioning of MI EEG-based BCIs can have severe consequences for individuals relying on these devices. In fact, suboptimal performance can lead to frustration, inaccuracy, and reduced functionality [10].

Hence, to enhance effectiveness, it is necessary to prioritize transparency in BCIs. This can result in the improved operational efficiency and smoother integration of BCI technology into daily life, ultimately enriching the quality of life for individuals with motor disabilities [11]. Still, the factors contributing to the limited usefulness of MI tasks are complex and varied. Intersubject variability is a noteworthy aspect that contributes to poor performance. In this sense, the subject mental state, attention, and fatigue can also substantially influence [12]. Additionally, the quality of electrical activity patterns generated by the brain plays a crucial role in controlling external devices [13]. However, these patterns exhibit substantial variation among subjects, even under identical stimuli or conditions [14]. Various factors, including gender, age, lifestyle, neurophysiological and psychological parameters, genetic differences, and cognitive processes, contribute to this variability [15]. Such diversities in brain patterns result in performance fluctuations, impeding the development of reliable and accurate BCIs [16].

Moreover, noise in EEG signals significantly contributes to this variability, obscuring underlying neural activity [17]. Notably, noisy records can originate from diverse sources, such as electromagnetic interferences, movement artifacts, individual skull thickness, and conductivity differences [18,19]. These unwanted signals make it difficult to identify the neural activity patterns that drive BCI performance accurately. Additionally, the need for interpretability in BCIs poses a critical challenge, hindering the identification of different patterns between high-performing and low-performing subjects. The difficulty in interpreting MI EEG-based BCIs and understanding their decision-making procedures complicates devising and enriching MI functionality [20].

In recent years, several methods have been proposed to enhance the performance of BCIs during the preprocessing and feature extraction stages. The preprocessing methods aim to mitigate the impact of low signal-to-noise ratio (SNR) caused by environmental and physiological artifacts, such as electrical noise, eye and muscle movements, heart activity, and respiration [21]. Additionally, the preprocessing stage seeks to tackle the low spatial resolution challenge caused by the volume conduction effect [22]. Additionally, artifacts in EEG signals can be removed using regression-based techniques, which use linear approaches to remove the noise [23]. Bandpass and notch filters can also eliminate electrical and environmental noise and frequency bands where neurophysiological information is irrelevant [24]. Blind source separation techniques, such as canonical correlation analysis (CCA), principal component analysis (PCA), and independent component analysis (ICA), are commonly used to decompose the contaminated EEG into statistically independent components to remove or correct the artifact [25]. Of note, ICA is recognized for its success in eliminating various types of artifacts [26]. Furthermore, different spatial filters have been proposed to overcome the volume conduction issue, including the common average reference (CAR) and the surface Laplacian (SL). The CAR spatial filter subtracts the average electrical activity measured across all sensors from each sensor to reduce the recorded noise [27]. Nevertheless, it does not address sensor-specific noise and may introduce noise into an otherwise clean sensor [28]. In contrast, SL aspires to remove the common brain activity of neighboring sensors due to the volume conduction effect, which improves local topographical features, facilitates sensor-level connectivity analysis, and helps to enhance the SNR [29]. Despite the effectiveness of these methods, applying them to all subjects regardless of the individual noise level can be detrimental to subjects with clean EEG [30].

On the other hand, the feature extraction strategies seek to transform the raw EEG signals into relevant brain patterns independent of subject-specific differences. This approach allows for identifying common patterns across individuals, improving the generalizability of BCI systems. Feature extraction techniques can be broadly categorized into time, time–frequency, and spatial approaches. In the time domain, amplitude modulation [31] and time domain analysis of variance [32] are widely used to extract features related to the amplitude and timing of specific EEG components, providing insights into the underlying neural processes involved in MI. These features enable the identification of significant differences between classes that can be used to classify the signals effectively. In the time–frequency

domain, wavelet transform [33] is a commonly used method that analyzes the changes in the frequency content of the EEG signal over time. This method provides information about the temporal dynamics of neural processes during MI, including evoked-related algorithms and intertrial coherence to capture the temporal evolution during the MI task [34]. Common spatial patterns (CSP) and FCs are standard methods for feature extraction in the spatial domain. CSP projects the EEG signals into a lower dimensional space using a set of learned spatial filters that enhance the differences between MI classes [35]. FCs capture the similarity between EEG channels, providing information on which brain regions interact when a subject performs the MI task [36]. However, choosing the appropriate feature extraction method for the MI task is challenging, as it demands considerable subject matter expertise and prior knowledge about the anticipated EEG signal [37]. Moreover, the specificity of the EEG signals' preprocessing steps for the interesting feature could exclude potentially relevant patterns from the analysis [38].

Nowadays, deep learning methods have emerged as a promising approach to overcoming the limitations of traditional methods in addressing MI intersubject variability by automating the preprocessing and extracting relevant features from EEG signals within an end-to-end framework [39]. In particular, models such as EEGNet, ShallowConvNet, Deep-ConvNet, graph convolution neural networks (GCN) [40,41], and EEG-transformer [42] have great potential to tackle EEG-based MI challenges. EEGNet and ShallowConvNet utilize convolutional layers to extract spatial and temporal patterns from EEG data. However, EEGNet may need help with capturing long-range temporal dependencies [43], while ShallowConvNet may not be as effective as deeper architectures in capturing complex patterns. DeepConvNet excels at capturing spatial and temporal patterns but requires much training data to avoid overfitting [44]. GCNs capture spatial relationships between electrodes by aggregating information from neighboring nodes in the graph. Nonetheless, they are sensitive to graph construction from EEG signals [45]. Recently, transformer-based models, such as EEG-transformer, have been adept at processing variable-length sequences by employing a self-attention mechanism to capture dependencies between different segments. Nonetheless, these models come with higher computational costs and require large amounts of samples [46].

Overall, deep learning can be prone to overfitting when they are too complex or the training data are noisy and insufficient [47]. Diverse regularization techniques have been proposed to tackle this issue. For example, domain adaptation aims to reduce variability across different subjects by learning a mapping between source and target spaces [48]. However, it requires a substantial amount of labeled data from both domains [49]. Multitask learning leverages information from related tasks to improve the performance of individual tasks [50]. Nevertheless, it assumes the availability of multiple related tasks, which may need to be more practical in specific scenarios [51]. Dropout and batch normalization are also helpful techniques that can reduce overfitting. The former randomly drops out a fraction of neurons during training to enhance the model's ability to learn robust features [52]. The latter normalizes input features across subjects to enhance network stability and convergence [53]. However, both techniques can increase computational requirements, and their performance can be sensitive to hyperparameter tuning and noisy samples [54,55]. FC-based regularizers introduce a penalty term to obtain low-rank or sparse connectivity matrices, reducing the impact of MI intersubject variability [28]. Regardless, these regularizers assume a smooth or sparse connectivity structure of the brain, which may not always hold in practice [56].

Here, we introduce a novel deep learning approach for EEG-based MI classification: kernel-based regularized EEGNet (KREEGNet). Our approach addresses the challenges posed by intrasubject variability in noisy EEG records and the lack of spatial interpretability in existing end-to-end frameworks used for MI classification. KREEGNet enhances the well-established EEGNet architecture, incorporating a twofold approach: (i) a kernel-based layer for Gaussian functional connectivity estimation is coupled within the EEGNet architecture, and (ii) a centered kernel alignment (CKA) loss is associated with a conventional cross-



entropy measure for deep learning classification to deal with noisy EEG records while preserving the spatial interpretability based on kernel mappings. Through experimentation on binary and multiclass MI classification databases, we demonstrate the superiority of KREEGNet over the baseline EEGNet and other state-of-the-art methods. Moreover, we explore the interpretability of our model at both individual and group levels, employing classification performance measures and pruned functional connectivities. Our findings highlight KREEGNet as a promising and interpretable deep learning approach for EEG-based BCI systems.

The agenda is as follows: Section 2 describes the materials and methods. Sections 3 and 4 present the experiments and discuss the results. Finally, Section 5 outlines the concluding remarks.

## 2. Materials and Methods

### 2.1. Centered Kernel Alignment Fundamentals

Let $X \subset \mathcal{X}$, $Y \subset \mathcal{Y}$ be a pair of random variables holding the samples $x \in X$ and $y \in Y$, respectively. The kernels $\kappa_X : \mathcal{X} \times \mathcal{X} \to \mathbb{R}$ and $\kappa_Y : \mathcal{Y} \times \mathcal{Y} \to \mathbb{R}$ can be defined to code nonlinear relationships among samples from positive definite functions, yielding

$$\kappa_X(x, x') = \langle \phi_X(x), \phi_X(x') \rangle_{\mathcal{H}_X}, \tag{1}$$

$$\kappa_Y(y, y') = \langle \phi_Y(y), \phi_Y(y') \rangle_{\mathcal{H}_Y}, \tag{2}$$

where $\phi_X : \mathcal{X} \to \mathcal{H}_X$ and $\phi_Y : \mathcal{Y} \to \mathcal{H}_Y$, with $\mathcal{H}_X$ and $\mathcal{H}_Y$ being the resulting reproducing kernel Hilbert spaces (RKHSs). Hence, the statistical alignment between $\kappa_X$ and $\kappa_Y$, $\rho_{CKA}(X, Y) \in [0, 1]$, referred to as centered kernel alignment (CKA), is calculated by taking the normalized inner product between them and averaging it across all pairs of realizations, as shown below [57,58]:

$$\rho_{CKA}(X, Y) = \frac{\mathbb{E}_{XY}\{\tilde{\kappa}_X(x, x')\tilde{\kappa}_Y(y, y')\}}{\sqrt{\mathbb{E}_X\{\tilde{\kappa}_X(x, x')\}\mathbb{E}_Y\{\tilde{\kappa}_Y(y, y')\}}}, \tag{3}$$

where $x, x' \in X$ and $y, y' \in Y$, $\mathbb{E}\{\cdot\}$ are the expectation operator, and $\tilde{\kappa}_Z$ stands for centered kernel aiming to provide translation invariance, as follows:

$$\tilde{\kappa}_Z(z, z') = \kappa_Z(z, z') - \mathbb{E}_z\{\kappa_Z(z, z')\} - \mathbb{E}_{z'}\{\kappa_Z(z, z')\} + \mathbb{E}_{zz'}\{\kappa_Z(z, z')\}, \tag{4}$$

which is defined for a given $Z \subset \mathcal{Z}$ with samples $z, z' \in Z$. In practical applications, when provided with a set of input–output pairs $\{x_n \in \mathbb{R}^P, y_n \in \mathbb{R}^Q\}_{n=1}^N$, we can compute the kernel matrices $K_X, K_Y \in \mathbb{R}^{N \times N}$ as $K_X[n, n'] = \kappa_X(x_n, x_{n'})$ and $K_Y[n, n'] = \kappa_Y(y_n, y_{n'})$. Utilizing Equations (3) and (4), we can calculate the empirical estimate for the CKA alignment $\hat{\rho}_{CKA}(K_X, K_Y) \in [0, 1]$:

$$\hat{\rho}_{CKA}(K_X, K_Y) = \frac{\langle \tilde{K}_X, \tilde{K}_Y \rangle_F}{\sqrt{\|\tilde{K}_X\|_F, \|\tilde{K}_Y\|_F}}, \tag{5}$$

where $\| \cdot \|_F$ and $\langle \cdot, \cdot \rangle_F$ are the Frobenius norm and inner product, respectively. Additionally, the centered kernel matrices in Equation (5) are calculated as $\tilde{K}_X = HK_XH$ and $\tilde{K}_Y = HK_YH$, with $H = I - \frac{1}{N}\mathbf{1}^\top\mathbf{1}$ ($I$ and $\mathbf{1}$ are the identity matrix and the all-one vector of proper size, respectively). As a result, the alignment described in Equation (5) serves as a data-driven estimator, enabling us to quantify the similarity between the random variables $X$ and $Y$.

### 2.2. Gaussian Functional Connectivity from EEG Records

Let us examine a collection of multichannel EEG recordings referred to as $\{X_n \in \mathbb{R}^{C \times T}\}_{n=1}^N$, where $C$ denotes the number of channels, $T$ represents the samples within EEG recordings, and $N$ is the number of trials.

Next, let us consider two EEG channels of a given trial $x_c, x_{c'} \in X$; with $c, c' \in \{1, 2, \ldots, C\}$, a pairwise correlation between the EEG channels can be computed as

$$\hat{\rho}_L(x_c, x_{c'}) = \frac{1}{T} \langle x_c, x_{c'} \rangle_2, \tag{6}$$

where $\langle \cdot, \cdot \rangle_2$ stands for the inner product. The pairwise linear relationships in Equation (6) allow for computing functional connections between EEG channels as an undirected graph representation.

However, we can effectively capture the nonlinear interactions among various channels by operating a generalized stationary kernel that transforms the input space into an RKHS. This approach enables us to obtain a more precise depiction of the underlying neural activity. Moreover, employing a stationary kernel guarantees that the proposed technique can effectively capture the temporal dynamics of EEG signals.

Given these considerations, the Gaussian kernel is widely preferred in pattern analysis and machine learning. It can approximate any function and offers mathematically tractable properties [59]. Therefore, it is an excellent choice for computing pairwise connections as a Gaussian-based functional connectivity (GFC) measure from the kernel function $\kappa_G : \mathbb{R}^T \times \mathbb{R}^T \to [0, 1]$, as [60]:

$$\kappa_G(x_c - x_{c'}; \gamma) = \exp\left(-\frac{1}{2}\gamma \|x_c - x_{c'}\|_2^2\right), \tag{7}$$

where $\|\cdot\|_2$ denotes the two-norm operator and $\gamma \in \mathbb{R}^+$ represents a scale parameter. The inclusion of a Gaussian function in Equation (7) facilitates the accurate and efficient calculation of the nonlinear interactions between $x_c$ and $x_{c'}$.

### 2.3. KREEGNet: Kernel-Based Regularized EEG Network

Let us consider an input–output set consisting of multichannel EEG records and labels denoted as $\{X_n \in \mathbb{R}^{C \times T}, y_n \in \{0, 1\}^Q\}_{n=1}^N$. Here, $y_n$ gathers the target labels for MI tasks encoded using the one-hot encoding (with $Q$ classes being considered). Our kernel-based regularized EEG network (KREEGNet), an enhanced version of the well-known EEGNet [61], enables accurate prediction of the MI label $\hat{y} \in [0, 1]^Q$ for a given EEG trial $X$. This prediction is accomplished through two primary blocks. Initially, the class membership prediction is performed as follows:

$$\hat{y} = (\varphi_Q \circ \varphi_T \circ \varphi_C \circ \varphi_{\breve{F}})(X), \tag{8}$$

where notation $\varphi(\tilde{X}) = \xi_\varphi(W_\varphi \otimes \tilde{X} + b_\varphi)$ stands for deep-learning-based layer mapping, $\circ$ is the function composition operator, and $\otimes$ is the tensor product, e.g., convolutional or fully connected layer-based operations. Additionally, $\tilde{X}$ is a given network's feature map of proper size, $W_\varphi, b_\varphi$ gather the weight matrix and bias vector of the layer, and $\xi(\cdot)$ is a nonlinear activation function. Namely, each layer function in Equation (8) is described as

- $\varphi_{\breve{F}} : \mathbb{R}^{C \times T} \to \mathbb{R}^{\breve{F}, C, T}$ is a convolutional layer holding $\breve{F}$ filters, a batch normalization, and a linear activation.
- $\varphi_C : \mathbb{R}^{\breve{F} \times C \times T} \to \mathbb{R}^{\alpha \breve{F}, C, \frac{T}{4}}$ is a depthwise convolutional layer holding ELU activation ($\alpha$ gathers the number of spatial filters), followed by an average pooling and a dropout operation.
- $\varphi_T : \mathbb{R}^{\alpha \breve{F}, C, \frac{T}{4}} \to \mathbb{R}^{\breve{F}', \frac{T}{32}}$ is a separable convolutional layer with ELU activation ($\breve{F}'$ is the number of pointwise filters), setting a batch normalization, an average pooling, and a dropout.
- $\varphi_Q : \mathbb{R}^{\breve{F}', \frac{T}{32}} \to [0, 1]^Q$ is a fully connected classification layer fixing a flatten operation and a softmax activation.

In turn, a kernel-based regularizer is applied by properly computing the data-driven GFC:

$$K^{\dagger} = (\widetilde{\kappa} \circ \varphi_{\check{F}})(X), \tag{9}$$

where $\varphi_{\check{F}}$ is defined as in Equation (8), $\widetilde{\kappa} : \mathbb{R}^{\check{F},C,T} \to [0,1]^{\check{F},C,C}$ extracts the GFC among EEG channels (see Equation (7)) along each of the $\check{F}$ filters, and $K^{\dagger} \in [0,1]^{\check{F},C,C}$.

Furthermore, the parameter set $\theta$, stacking the weight matrices and bias vectors in Equation (8), and the scale parameter $\gamma$ of the GFC in Equations (7) and (9), is optimized using a gradient descent-based framework with back-propagation [62]:

$$\theta^* = \arg \min_{\theta} \frac{(1-\lambda)}{N} \sum_{n=1}^{N} CE(y_n, \hat{y}_n(\theta)) - \frac{\lambda}{\check{F}} \sum_{f=1}^{\check{F}} \hat{\rho}_{CKA}(\check{K}_f(\theta), K_\delta), \tag{10}$$

where $\lambda \in [0,1]$ is a trade-off hyperparameter and $CE(\cdot, \cdot)$ stands for the cross-entropy loss defined as

$$CE(y_n, \hat{y}_n(\theta)) = - \sum_{q=1}^{Q} y_{nq} \log(\hat{y}_{nq}(\theta)), \tag{11}$$

with $y_{nq} \in y_n$ and $\hat{y}_{nq}(\theta) \in \hat{y}_n$. Moreover, the kernel-matrix $\check{K}_f(\theta) \in [0,1]^{N \times N}$ is computed as

$$\check{K}_f[n, n'] = \left\langle \text{triu}(K_n^{\dagger}(\theta; f)), \text{triu}(K_{n'}^{\dagger}(\theta; f)) \right\rangle_2 \tag{12}$$

where $\text{triu}(K^{\dagger}(\theta; f)) \in [0,1]^{C\frac{(C-1)}{2}}$ holds the upper triangular matrix of the GFC stored in $K^{\dagger}$ for filter $f$. Likewise, the target kernel matrix $K_\delta \in \{0,1\}^{N \times N}$ is built as

$$K_\delta[n, n'] = \delta(|y_n - y_{n'}|_1), \tag{13}$$

with $\delta(\cdot)$ being the delta function and $|\cdot|_1$ the 1-norm.

The optimization problem outlined in Equation (10) enables the training of our KREEG-Net for MI discrimination. Figure 1 summarizes the KREEGNet pipeline. To ensure numerical stability and simplicity, the GFC scale parameter is learned as a mapping of $10^\gamma$. It is worth mentioning that a preprocessing stage is included to align the various database conditions, such as sample frequency, bandpass filtering, and EEG window size.

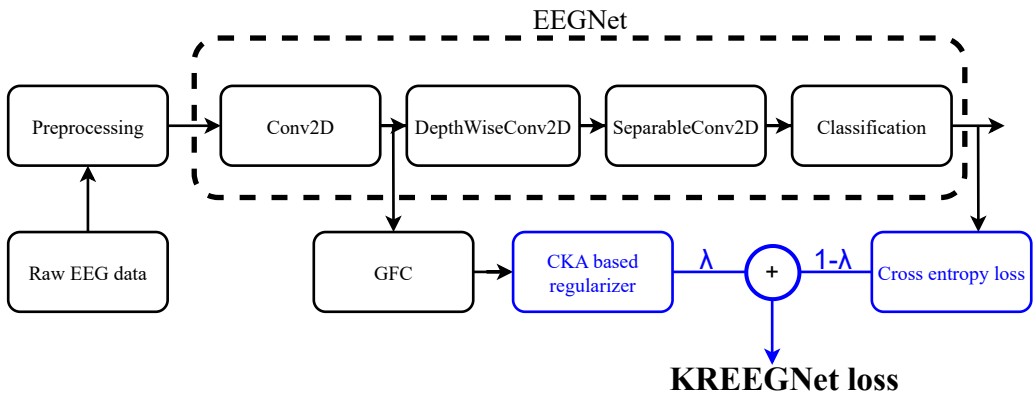

**Figure 1.** KREEGNet pipeline for motor imagery classification from EEG records.

### 2.4. Group Analysis from EEGNet and KREEGNet Performance

We construct a scoring matrix for robust validation with rows equivalent to the MI subjects' performance, gathering six columns representing accuracy, Cohen's kappa, the area under the ROC curve scores, and their corresponding standard deviations. To maintain the principle of 'the higher, the better' and restrict all column values within the $[0,1]$ range,

we substitute the standard deviation with its complement and normalize Cohen's kappa by adding one and dividing by two.

Following that, we utilize this scoring matrix and the k-means clustering algorithm [59], setting *k* to three, to train a model that categorizes subject results based on the benchmark model EEGNet [61] into three groups: top performers (GI), average performers (GII), and low performers (GIII). Subsequently, our KREEGNet's subject analysis results are clustered using the trained *k*-means and the score matrix. The ultimate goal is to examine and discern how subject classification shifts between the EEGNet and the KREEGNet-based groups [60].

## 3. Experimental Setup

We provide a comprehensive overview of the pipeline used to develop and evaluate the KREEGNet model for motor imagery discrimination. It includes an analysis of the datasets used, the training phase of the model, and the techniques employed to assess the proposal's effectiveness.

### 3.1. Dataset Description

In order to evaluate the effectiveness of our KREEGNet, we conducted tests on two well-known databases that involve motor-related tasks.

**DBI: BCI Competition 2008**—Graz Dataset 2a (http://www.bbci.de/competition/iv/index.html , accessed on 1 April 2023). The dataset contains EEG data from nine subjects who participated in a motor imagery paradigm consisting of four tasks: imagining the movement of the left hand, right hand, both feet, and tongue (four-class problem). The data were collected in two sessions on different days, comprising six runs with 48 trials per run (12 for each class). This resulted in a total of 288 trials per session. A short acoustic warning and a cross on a black screen signaled the start of each trial, which lasted 7 s. Then, at 2 s, a visual cue appeared on the screen for 1.25 s, indicating which MI task to perform until the cross disappeared at 6 s. Next, a short break followed, and the screen went black. The EEG data were recorded using a 22-channel Ag/AgCl electrode montage based on the 10/20 system. In addition, three EOG electrodes were also used to record ocular artifacts. The signals were sampled at 250 Hz and bandpass-filtered between 0.5 and 100 Hz, with a 50 Hz notch filter applied. The datasets for each subject and session were stored in the general data format for biomedical signals, with one file per subject and session.

**DBII: GiGaScience** (http://gigadb.org/dataset/100295, accessed on 1 April 2023). It includes EEG data from 52 healthy subjects, although only 50 are available for evaluation. The data were acquired in one session using the MI experimental paradigm with two classes (left and right hands). Each session comprised five or six runs with 100 or 120 trials per class. Moreover, each trial lasted 7 s, starting with a black screen with a fixation cross within 2 s. A cue instruction appeared randomly on the screen within 3 s, prompting the subject to perform the indicated MI task. The trial ended with a blank screen and a 4.1 to 4.8 s break. The EEG data were collected using a Biosemi ActiveTwo system with 64 Ag/AgCl electrodes placed according to the 10/10 international system, sampled at 512 Hz, and stored in *.mat format. Actual left-hand and right-hand movements and six types of noise (blinking eyes, eyeball movement up/down, eyeball movement left/right, head movement, jaw clenching, and resting state) were also collected, aside from the MI recordings.

Figure 2 provides an overview of the DBI (four-class problem) and DBII (binary-class problem) montage and paradigm used for MI classification.

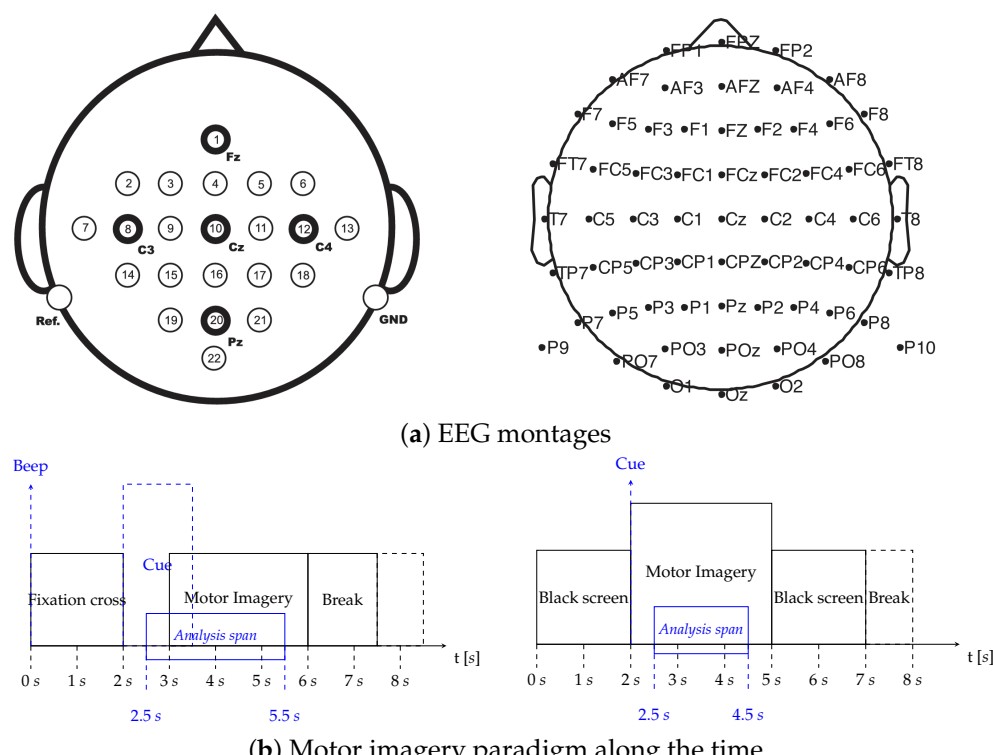

**(a)** EEG montages

**(b)** Motor imagery paradigm along the time

**Figure 2.** The EEG-MI databases examined: DBI (BCI competition four-class task) and DBII (Giga-Science binary task), displayed in the left and right columns, respectively. The **top row** shows the EEG montages, while the **bottom row** presents the MI paradigm tested.

### 3.2. KREEGNet Training Details and Assessment

The training of our KREEGNet comprises three stages: (i) preprocessing of EEG records, (ii) fine-tuning the network hyperparameters for improved classification performance, and (iii) interpreting functional connectivity by identifying relevant patterns learned during deep learning training.

To initiate EEG preprocessing, a custom database loader module (see https://github.com/UN-GCPDS/python-gcpds.databases, accessed on 8 April 2023) was utilized to load the recordings. Only EEG channels were considered, and the signals were scaled to $\mu V$ to ensure suitability for analysis. Any trials marked as bad were rejected. A fifth-order Butterworth bandpass filter was applied to all channels within the $[4, 40] H_z$ range, where MI activity was observed [60]. Additionally, each channel's signal was clipped within the postcue onset time window, retaining only information from the MI task. For DBI, the time window was 0.5–3.5 s, while for DBII, it was 0.5–2.5 s. Then, to ensure that the network parameters remained consistent, each channel's signal was downsampled in both databases from 256 Hz for DBI and 512 Hz for DBII to 128 Hz. Our preprocessing step is similar to the one described by the authors in [61].

Next, to ensure a reliable evaluation of our model, we employed the stratified shuffle split fivefold 80–20 scheme within each subject's data. This process involved shuffling the data and selecting 80% for training while holding out the remaining 20% for testing. This procedure was repeated five times. Model performance was evaluated using accuracy, Cohen's kappa, and the area under the curve ROC [59]. An exhaustive search strategy for hyperparameter tuning was implemented, and the mean accuracy score across the folds was used to evaluate each hyperparameter's performance. In order to train our model, we formulated the loss function as a combination of the cross-entropy (CE) and the CKA-based regularization, with each component weighted accordingly. The CE component served as a guide for the model to perform the classification task effectively. On the other hand, the CKA component played a role in mitigating overfitting by considering the spatial information of the FCs computed in the GFC layer. The contribution of each

term in the cost function was defined as $(1 - \lambda)$ for the CE component and $\lambda$ for the CKA component (see Equation (10)). The value of $\lambda$, a hyperparameter, was searched within the set $\{0, 0.2, 0.4, 0.6, 0.8\}$. We employed the Adam optimizer with an initial learning rate of $1 \times 10^{-3}$ to optimize the network parameters. Additionally, a callback mechanism was implemented to decrease the learning rate by 10 when the loss function no longer exhibited improvement. The KREEGNet was trained for 500 epochs, utilizing all available samples in the training set.

The experiments conducted in this study were performed using Python version 3.8 in both Google Collaboratory and Kaggle environments. We employed TensorFlow version 2.8.2 to construct models, define losses, create custom layers, and implement training strategies. To ensure reproducibility and facilitate further analysis and experimentation, we consistently saved the model weights and performance scores. For those interested in reproducing the training of our KREEGNet, we have provided a Kaggle notebook accessible at the following link: https://www.kaggle.com/mateotobonhenao/kreegnet-training (accessed on 8 April 2023). This resource contains all the necessary details and code to replicate our training procedure.

*3.3. Method Comparison*

To assess the efficacy of our KREEGNet, we conducted a comprehensive analysis of its classification performance and the discriminability of estimated FCs. Additionally, we categorized subjects into groups (for DBII) based on their classification performance to gain insights into the impact of our proposal against four classical end-to-end deep learning models that incorporate both temporal and spatial information from EEG signals using stacked 1D convolutions. The first model, the baseline EEGNet [61], utilizes separable convolutions to reduce parameters while maintaining performance similar to traditional convolutional layers. In addition, it includes a depthwise convolution layer to capture spatial information and a fully connected layer with softmax activation for classification. The second model, ShallowConvNet [63], is a simpler architecture consisting of a single convolutional layer, followed by nonlinear activation, batch normalization, and pooling layers. Despite its simplicity, it effectively classifies EEG signals. The third model, Deep-ConvNet [63], is a deeper architecture comprising five convolutional layers, followed by nonlinear activation, batch normalization, and pooling layers. Although it performs well in EEG signal classification, it is computationally more expensive than ShallowConvNet and EEGNet. Finally, we consider the TCFussionnet proposed in [64]. This model consists of three main components: a temporal component that learns various bandpass frequencies, a depthwise separable convolution that extracts spatial features for each temporal filter, and a temporal convolutional network (TCN) block that captures temporal features. These features are combined to generate comprehensive feature maps, which are then classified into different MI classes using a dense layer with softmax activation. The Kaggle notebook available at https://www.kaggle.com/mateotobonhenao/dl-methods-comparison (accessed on 8 April 2023) contains the code necessary to assess the MI classification effectiveness of the aforementioned deep learning models. Additionally, the following GitHub repository holds the complete codes related to our experiments ( https://github.com/mtobonh/KREEGNet, accessed on 8 April 2023).

## 4. Results and Discussion

*4.1. Baseline EEGNet vs. KREEGNet: Subject and Group-Level Results*

We conduct a comparative analysis of KREEGNet with the widely recognized benchmark, EEGNet, for both DBI and DBII in the context of binary MI classification tasks, explicitly focusing on distinguishing between left- and right-hand imagery movements. A subject-specific examination is executed across both databases, while the group-level analysis (see Section 2.4) is limited solely to DBII due to DBI's composition of a mere nine subjects.

Figure 3a,b present a comparative accuracy analysis of subject-specific and group-level analysis. The dotted orange line in the figures corresponds to the EEGNet; in contrast, the dotted blue line illustrates the proposed KREEGNet. The blue and red bars in the figures indicate the impact of employing the KREEGNet on individual subject accuracy. Specifically, the blue bars denote improvements in accuracy, while the red bars indicate decreases. These visual cues provide valuable insights into the performance enhancements achieved by our approach across specific subjects. Moreover, in the context of DBII, the figure's background incorporates bars with low opacity in opal green, lemon yellow, and salmon pink. These color-coded backgrounds denote the grouping of subjects into top-performing, average-performing, and low-performing subjects.

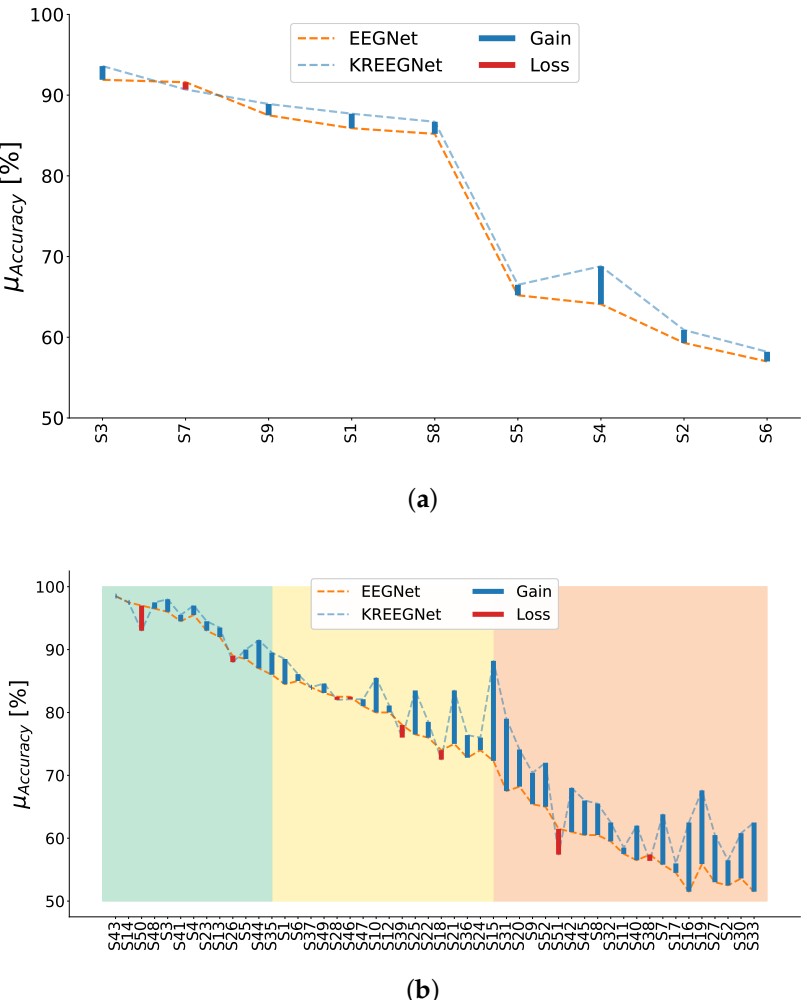

(**a**)

(**b**)

**Figure 3.** EEGNet vs. KREEGNet comparison results. The top row demonstrates the subject-specific analysis for DBI, while the lower row exhibits the group-level evaluation for DBII (KREEGNet gain: GI 1.0%, GII 2.9%, and GIII 5.7%). The reported mean accuracy corresponds to a binary MI classification of left- versus right-hand movement. Subjects have been organized following their EEGNet performance. The blue bars signify an enhanced performance achieved by our proposed KREEGNet, whereas the red bars highlight instances of reduced performance. The backdrop for the DBII results visually represents the group membership, with top performers in GI, average performers in GII, and low performers in GIII. (**a**) Average accuracy: EEGNet 76.4%, KREEGNet A 78.0%. (**b**) Average accuracy: EEGNet 74.4%, KREEGNet A 77.9%.

Our KREEGNet model's performance regarding DBI reveals a subject-dependent average accuracy of 78.0%, surpassing the baseline EEGNet by 1.6%. Notably, out of all the subjects, only Subject 7 (S7) experienced a marginal decrease in performance, with a

decline of less than 1%. Conversely, the remaining subjects demonstrated improvements in accuracy. Subject 4 (S4) was particularly impressive, exhibiting a remarkable performance increase of 4.7%, showcasing the effectiveness of our KREEGNet model in enhancing subject-specific analysis by coding relevant functional connections among channels within an end-to-end regularized network.

For DBII, the EEGNet and KREEGNet models achieved subject-dependent average accuracies of 74.4% and 77.9%, indicating an improvement of 3.5% for our proposal. The standard deviations for EEGNet and KREEGNet were 14.9% and 13.2%, respectively, suggesting that our approach resulted in less variability among subjects' performance. Interestingly, the accuracy of KREEGNet varied across the subjects, with three scenarios emerging from the results. First, 8 subjects showed a decrease in accuracy, with only 3 experiencing a reduction of 2% or more. Second, 2 subjects did not show any change in accuracy. Lastly, the remaining subjects demonstrated an increase in accuracy, with 19 of them experiencing an increase of more than 5%.

Now, the impact of our method on the performance of different subject groups in DBII was substantial. In the case of Group GIII, KREEGNet outperformed the baseline in all but two instances, with a remarkable increase of over 5% observed in 14 cases. As for Group GII, 4 subjects experienced a minor decrease of less than 2%, while 1 remained unchanged. On the other hand, 12 subjects showed a performance improvement, with half achieving an increase of over 3%. Of particular note was Subject 15, which exhibited an impressive performance boost of 16%, highlighting the strong influence of our CKA-based regularizer on specific individuals. In Group G III, only 2 subjects witnessed a decrease in accuracy, while 9 subjects demonstrated improved performance, including 2 with increases exceeding 3%. So then, our strategy yielded significant performance enhancements for most subjects across all groups, with a notable benefit observed in the poorly performing subject group.

Similarly, Figure 4 presents the categorization of the subject group and the influence of KREEGNet. The initial row displays the arrangement of subjects as per the results of EEGNet, while the final row illustrates the shift or constancy of each subject's group derived from the KREEGNet outcomes. For example, in GIII, our approach promoted 4 subjects to GII. Likewise, 2 individuals were elevated from GII to GI. Importantly, no individual experienced an in-group demotion status, underlining the equal or superior performance of KREEGNet compared with the standard EEGNet.

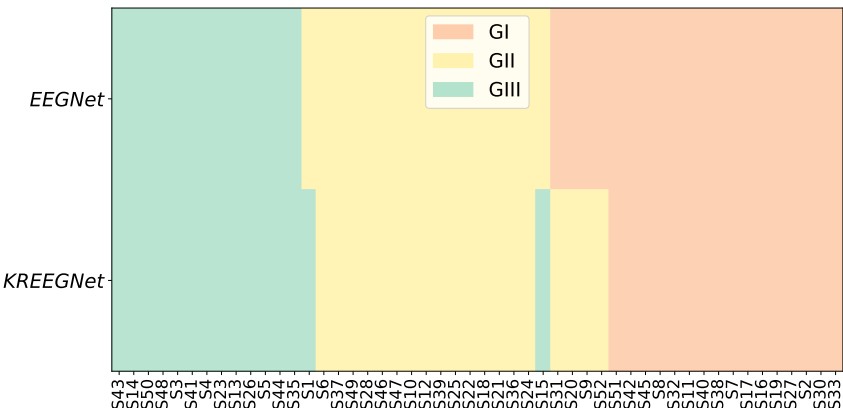

**Figure 4.** KREEGNet subject group enhancement (baseline: EEGNet). Note that green, yellow, and red represent top, average, and low performance regarding the average accuracy along subjects. **First row**: The arrangement of subjects according to EEGNet classification. **Second row**: Alterations in subject group affiliations based on the results of KREEGNet.

Subsequently, we scrutinized the complex behavior of the hyperparameters $\lambda$ and $\gamma$ across different subject groups in DBII. $\lambda$ symbolizes the importance given to the CKA-

based regularizer in the cost function of KREEGNet, contributing to enhancing the network's classification capabilities. Conversely, $\gamma$ sets the bandwidth scale for the Gaussian kernel employed in the GFC layer that calculates the FCs. By investigating the dynamics of these hyperparameters, we seek to understand their influence on performance and the GFC layer's FC estimation. Figure 5a presents a boxplot depicting the statistical distribution of the $\lambda$ hyperparameter among the subject groups, with the background boxes denoting group membership. First, most tend to possess lower $\lambda$ values in GI, specifically below 0.6. This is attributed to the fact that subjects within this group display more evident MI patterns, readily captured by the standard EEGNet model. Second, GII exhibits a more evenly distributed set of values, with half of the subjects presenting $\lambda$ values exceeding 0.3. This could imply that some subjects at this stage demonstrate noisy MI patterns that heighten the risk of overfitting the training data, thereby reducing the classification performance. Lastly, for GIII, $\lambda$ values are predominantly higher. Precisely, half of the subjects in this group have $\lambda$ values above 0.5, with the majority of the remainder having values ranging between 0.4 and 0.5. The latter suggests that most of the subjects' data in this group present noisy patterns. Nevertheless, the CKA-based regularizer, working on the FCs computed by the GFC layer, aids in eliminating this unwanted effect, leading to improved classification performance.

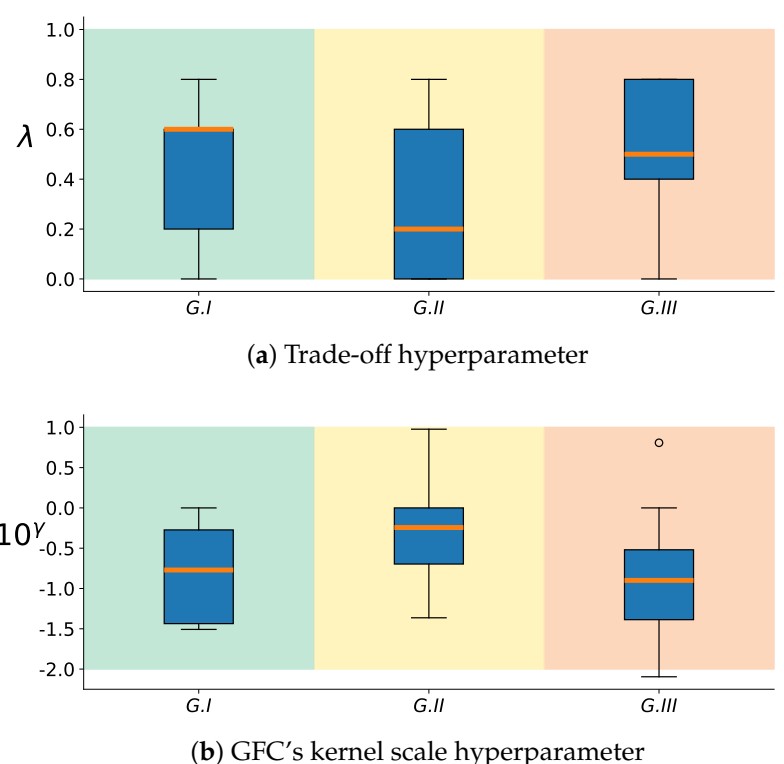

(**a**) Trade-off hyperparameter

(**b**) GFC's kernel scale hyperparameter

**Figure 5.** Analysis of KREEGNet hyperparameters at the group level for DBII. Boxplot diagrams are provided for the tuned $\lambda$ and $\gamma$ values in relation to the top- (GI), average- (GII), and low- (GIII) performing subjects.

In the same way, Figure 5b displays the boxplot of the $\gamma$ hyperparameter among different subject groups. This bandwidth filters the relationships between channels, suggesting that channels with higher noise levels have lower bandwidth values to circumvent unwarranted connections. The findings imply that subjects in GIII require more filtering through the $\gamma$ parameter, hinting that these individuals typically have higher noise in their MI patterns. Our CKA-based regularizer and the GFC layer contribute to the reduction of these noises, thereby enhancing classification performance. Notably, our results demonstrate an inverse linear relationship between the fixed $\lambda$ and $\gamma$ values. Specifically, subjects with good performance, i.e., those in G I and some in G II, exhibit lower values of $\lambda$ and

higher values of $\gamma$, indicating a low contribution of the CKA-based regularizer and that the bandwidth of the GFC layer is more flexible in filtering out the relationship between channels. This suggests that the MI patterns for these subjects are cleaner and less affected by noise. Contrariwise, subjects with poor performance, i.e., those in G III, exhibit higher values of $\lambda$ and lower values of $\gamma$, indicating that the CKA-based regularizer contributes more to the cost function to reduce the effect of overfitting due to the presence of noise. Additionally, $\gamma$ shrinks the value of the bandwidth in the GFC layer to be more rigid in filtering out the relationship between channels, thereby avoiding spurious connectivities. These findings highlight our KREEGNet's importance in optimizing the performance and interpretability of EEG-based MI tasks.

### 4.2. Relevance Analysis Results

We evaluated the FC variations across subjects, focusing on determining which connections significantly influence the ability to distinguish between the MI classes. Acknowledging that a strong correlation in the FC matrix does not automatically translate into enhanced class distinction is essential. In this endeavor, we utilized the Kolmogorov–Smirnov (KS) statistic [65], a tool that quantifies the disparity between the class distributions for each FC. Our KS-based connectivity pruning is as follows:

- We categorized each connection's trials for an individual based on the label, forming the right and left sample sets.
- Following this, we calculated the KS statistic for the connectivity between each pair of EEG channels along the training set trials. A KS value nearing 1 signifies a high level of distinguishability for the connectivity between two channels, whereas a value approaching 0 suggests a low level of separability. Here, the two-sample KS test compares the underlying distributions of two independent samples regarding the MI classes.
- Moreover, we utilized the maximum operator across the estimated feature maps to establish a KS statistic matrix. This matrix denotes the class separability of each connectivity.
- In order to illustrate the variations in each KS statistic matrix across subjects and groups, we depicted each matrix of KS statistic values on a two-dimensional scatter representation. Both dimensions were calculated employing the widely accepted *t*-SNE algorithm [66].
- Lastly, to fully comprehend the key connectivities and channels involved in the MI classification, we used topoplots from the KS statistic matrix.

Figures 6 and 7 depict the *t*-SNE 2D projections of the KS statistic matrices of each subject for DBI and DBII, respectively. In particular, the color-coded outer square of Figure 7 represents the group affiliation (GI, GII, and GIII). This visual representation enhances our comprehension of the significant connectivity patterns in the MI classification task. Figure 6 depicts the optimal performing subjects at the bottom, intermediate performers towards the left middle, and poorly performing ones at the top left. Notably, the KS statistic matrices of high-performing subjects are more distinct, except for Subject 7. This finding suggests that the FCs estimated by the GFC layer hold more significance in the MI classification. On the contrary, intermediate and poor performers show sparse KS matrices, implying that their data have a higher noise level, which results in erroneous FCs that overlap with MI class distributions.

Likewise, Figure 7 shows how G III exhibits sparse KS statistic matrices in the bottom-right corner, indicating that the FCs estimated are not discriminative among classes. This observation can be explained by the fact that the $\gamma$ parameter took lower values for this particular group of subjects, which tend to produce sparse matrices regardless of MI classes. In contrast, the subjects in G I in the top left tend to have more fired KS statistics, with a notable concentration over the MI area. Finally, G II reveals more erratic behavior, with the subjects near G III. The latter may be attributed to individual differences in brain activity during the MI tasks.

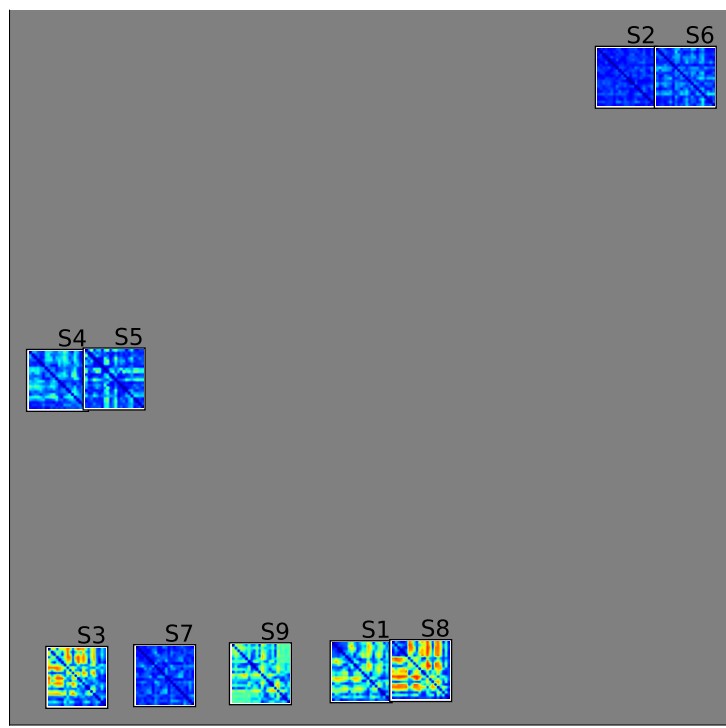

**Figure 6.** DBI-2D *t*-SNE projection of KS-based pruned FC matrices utilizing our KREEGNet. A gradation of colors ranging from blue to red represents a continuum from low to high separability.

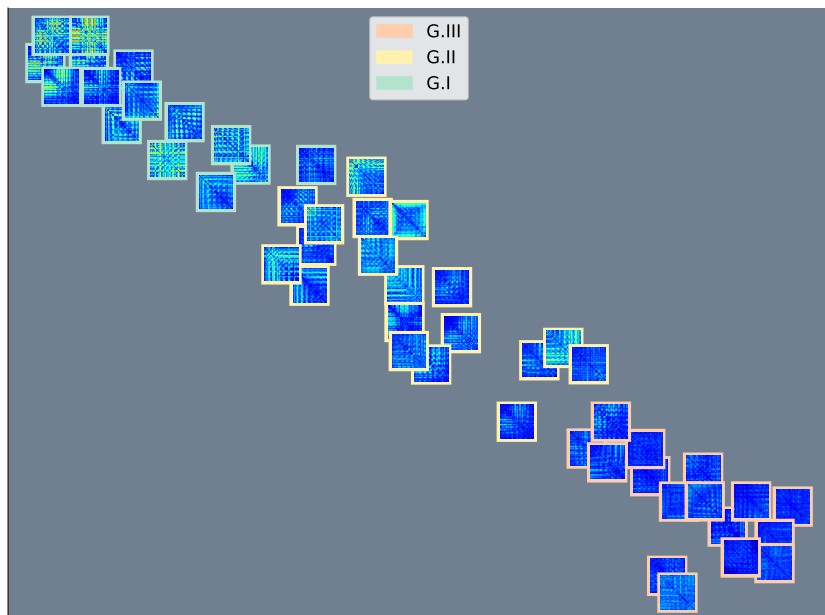

**Figure 7.** DBII-2D *t*-SNE projection of KS-based pruned FC matrices utilizing our KREEGNet. A gradation of colors ranging from blue to red represents a continuum from low to high separability. Outer boxes indicate subject group belongingness: green G I, yellow G II, and red G III.

In order to evaluate the informational dynamics of pruned FCs, we utilized quadratic Rényi's entropy, computed over the KS statistic matrices [67]. Our observations suggested that sparse KS statistic matrices corresponded with higher noise levels, whereas the KS matrices that had been freed up corresponded to lower noise levels. These statements are corroborated by Figure 8a,b. Furthermore, our findings align with the previously stated remarks. Specifically, the subjects that perform poorly in DBI tend to display higher entropy values, which is also true for the subjects classified under G III in DBII.

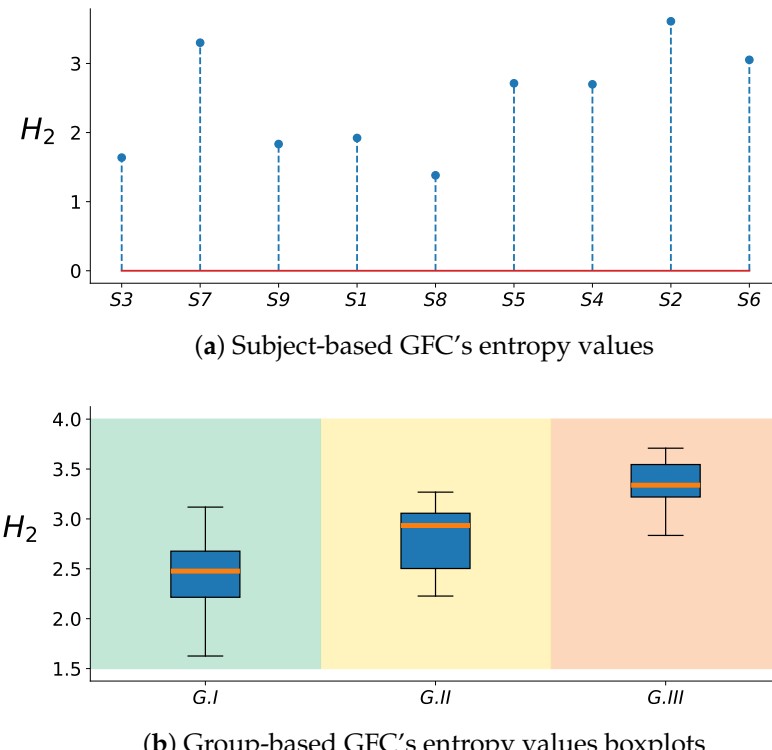

(**a**) Subject-based GFC's entropy values

(**b**) Group-based GFC's entropy values boxplots

**Figure 8.** Rényi's entropy-based retained information within the estimated functional connectivity matrices ($H_2$ stands for quadratic entropy value). **Top**: DBI results sorted regarding the classification performance. **Bottom**: DBII results where the background codes the group membership (best-, medium-, and poor-performing subjects). Boxplot representation is used to present the retained information within each group.

Next, the topoplots in Figure 9a,b show the distribution of relevant connectivities and channels. Relevant subjects are selected for visualization purposes in DBI. Meanwhile, the centroid of each group in DBII is employed. The results indicate that the sensorimotor area is the most critical region for both databases. It suggests that our KREEGNet effectively improves classification performance and model interpretability by incorporating a CKA-based regularizer and a *GFC* layer.

For DBI, we analyzed S3, S4, and S6 to represent high-performing, intermediate, and low-performing subjects. Notably, S3 exhibited a higher number of relevant FCs compared with S4 and S6. Furthermore, the FCs of S3 and S4 are thickened in the central brain region, consistent with the MI paradigm. However, S6 displayed a concentration of FCs in a single channel in the left-central region. Concerning DBII, the analysis of connectivities and channels in G I subjects revealed that the primary areas of interaction during MI tasks are located in the left-right central regions. This finding suggests that these subjects exhibit more distinct and reliable patterns of MI activity. The subjects belonging to G II displayed a pattern of connectivities and channels in the right-central brain region. However, a diffuse pattern was observed in the left hemisphere, covering some posterior and brain regions not strongly associated with MI activity. This diffuse pattern may be attributed to noise-induced EEG features, affecting classification performance. Finally, for the subjects in G III, the connectivities are concentrated in the central region of both hemispheres, which aligns with the MI paradigm. Similar to G II, the main channels are located in the right-left central brain areas, but robust patterns are observed in the left-posterior and frontal areas, highlighting noisy behavior.

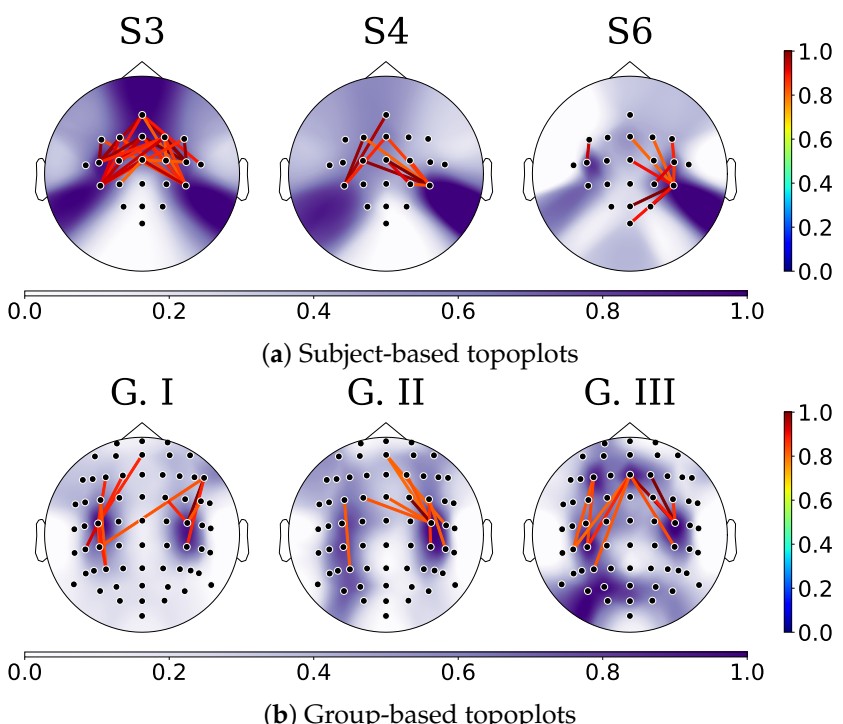

(**a**) Subject-based topoplots

(**b**) Group-based topoplots

**Figure 9.** Visual outcomes of the topographical maps (DBI and DBII results). The **top row** illustrates the results related to significant subjects for the DBI. The **bottom row** displays group-oriented visualizations for the DBII. Only those connections that hold a value surpassing the 95th percentile are highlighted. The backdrop of these visualizations corresponds to the normalized cumulative connection strength across channels, which is projected onto the topographical map.

### 4.3. Method Comparison Results: Binary and Multiclass MI Classification

The classification performances of the deep learning models discussed in Section 3.3 for DBI and DBII are presented in Tables 1 and 2, respectively. The results indicate that the DeepConvNet model performs the worst for both databases, while our proposed KREEGNet achieves the highest MI classification results. Notably, the ShallowConvNet, EEGNet, and TCFussionnet networks conduct similarly in both databases. Our KREEGNet attains outstanding results in all classification measures for DBII, demonstrating its superior performance. Although our model also achieves the best results for DBI, the difference in performance compared with other models is less significant. This can be attributed to the fact that DBI has fewer channels, with most of them concentrated in the central brain area, which limits the effect of the estimated FC by the GFC layer and the CKA-based regularizer. Then, only interactions between channels located in the same brain region are considered, reducing the diversity of information.

**Table 1.** Multiclass MI classification results for DBI. Average accuracy, kappa, and AUC are displayed $\pm$ the standard deviation.

| Approach | Accuracy | Kappa | AUC |
|---|---|---|---|
| DeepConvNet [63] | $55.5 \pm 24.3$ | $40.6 \pm 32.4$ | $78.1 \pm 20.4$ |
| ShallowConvNet [63] | $74.9 \pm 13.3$ | $66.7 \pm 17.7$ | $91.6 \pm 7.2$ |
| EEGNet [61] | $76.4 \pm 14.6$ | $68.6 \pm 19.5$ | $92.5 \pm 0.71$ |
| TCFussionnet [64] | $77.3 \pm 13.4$ | $69.7 \pm 17.9$ | $92.6 \pm 0.68$ |
| KREEGNet (ours) | $78.0 \pm 14.1$ | $70.7 \pm 18.8$ | $92.6 \pm 0.7$ |

**Table 2.** Binary MI classification results for DBII. Average Accuracy, Kappa, and AUC are displayed ± the standard deviation.

| Approach | Accuracy | Kappa | AUC |
|---|---|---|---|
| DeepConvNet [63] | 61.9 ± 12.4 | 23.6 ± 24.9 | 66.0 ± 16.0 |
| ShallowConvNet [63] | 72.5 ± 14.1 | 44.6 ± 28.3 | 77.9 ± 15.3 |
| TCFussionnet [64] | 73.9 ± 14.8 | 48.0 ± 30.0 | 80.0 ± 16.3 |
| EEGNet [61] | 74.4 ± 14.9 | 48.6 ± 29.8 | 79.6 ± 16.4 |
| KREEGNet (ours) | 77.9 ± 13.2 | 55.7 ± 26.5 | 82.5 ± 14.5 |

## 5. Conclusions

We introduced a novel deep learning approach for EEG-based motor imagery classification, named kernel-based regularized EEGNet, grounded on Gaussian functional connectivity and centered kernel alignment (KREEGNet). Our proposal addresses the issues of intrasubject variability caused by noisy EEG recordings and the absence of spatial interpretability in end-to-end frameworks for MI classification. Specifically, we amplified the widely recognized EEGNet architecture with a kernel-based layer designed to encode discriminant functional connectivities through a Gaussian similarity layer. Furthermore, the regularizer rooted in centered kernel alignment seeks to minimize the overfitting effect caused by noise in the EEG data of subjects, thereby enhancing the performance of motor imagery classification.

The experimental outcomes obtained from binary and multiclass EEG-based motor imagery classification databases revealed the superior performance of our KREEGNet compared with the baseline EEGNet and other state-of-the-art deep learning models. We further delved into the interpretability of our model at both the subject-dependent and group levels, using classification performance measures and pruned functional connectivities specific to our KREEGNet. In classifying subjects into three groups based on their performance, we showcased the capacity of KREEGNet to improve the MI classification performance, particularly among those subjects with previously poor performance. This highlights the fact that these individuals are most significantly impacted by noise.

In summary, the proposed KREEGNet effectively resolves issues such as intrasubject variability attributed to noise in EEG data and the absence of spatial interpretability in deep learning models used for motor imagery classification. These insights will aid in developing brain–computer interfaces that are more accurate and interpretable, expanding their potential for diverse applications.

In future research, we aim to augment our KREEGNet to achieve end-to-end functional connectivity estimation via graph convolutional networks [68]. Additionally, we intend to investigate causal connectivity rooted in information-theoretic learning for deep-learning-based estimations [69]. Lastly, we consider conducting subject-independent experiments and testing transformer networks [70].

**Author Contributions:** Conceptualization, A.M.Á.-M. and C.G.C.-D.; methodology, M.T.-H., A.M.Á.-M. and C.G.C.-D.; software, M.T.-H.; validation, M.T.-H. and A.M.Á.-M.; formal analysis, A.M.Á.-M. and C.G.C.-D.; investigation, M.T.-H.; data curation, M.T.-H.; writing—original draft preparation, M.T.-H. and C.G.C.-D.; writing—review and editing, A.M.Á.-M. and C.G.C.-D.; visualization, M.T.-H.; supervision, A.M.Á.-M. and C.G.C.-D. All authors have read and agreed to the published version of the manuscript.

**Funding:** Under grants provided by the project "Sistema de integración de EEG, ECG y SpO2 para seguimiento de neonatos en unidad de cuidados intensivos del Hospital Universitario de Caldas—SES HUC"—HERMES 57414, funded by Universidad Nacional de Colombia.

**Data Availability Statement:** Publicly available datasets were analyzed in this study. These data can be found here: https://www.bbci.de/competition/iv/ and http://gigadb.org/dataset/100295.

**Conflicts of Interest:** The authors declare no conflict of interest.

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
