# Peer review of "Kernel-Based Regularized EEGNet Using Centered Alignment and Gaussian Connectivity for Motor Imagery Discrimination"

_computers, doi:10.3390/computers12070145_

Round 1

Reviewer 1 Report

This paper deals with EEG-based motor Imagery classification using a method named Kernel-based Regularized EEGNet grounded on Gaussian Functional Connectivity and Centered Kernel Alignment (KREEGNet)

The paper is sound and the research area is relevant and of quite a potential interest for readers. It provides a solid theoretical background together with a clear presentation of the method used, followed by well-discussed results. Additionally, the results obtained using the new method were thoroughly compared to some alternative approaches.

Presentation standard does not leave much to be desired, tables are fine, some figures have e.g. axes description which are not readable (I would suggest improving this because otherwise the results presented in the figure may be difficult to comprehend - e.g. see Fig. Fig.3b -x asis description or Fig.4). 

The paper is written in a good English, if I were picky I could point out a few issues which do not affect the overall paper quality, e.g.:

- line 26 - "Acquiring brain activity" - I think that we can acquire signals reflecting brain activity,

- line 36 - "aiming to" - I believe the right form is "aiming at"

But other than that (and just a few similar in other few places) I really can't criticise use of English.

I have looked through the Authors' publications and some papers relevant to this domain and I think that the presented approach contains the novelty element so it may be an interesting read to the other researchers. Hence, I recommend this paper for publication just with a few little touches on the presentation standard and use of English side. No additional review on my end is needed.

Language does not much to be desired. I would suggest just one more go to get rid of some very minor issues (indicated above), but even in the current form it is comprehensible.

Reviewer 2 Report

The paper introduces the Kernel-based Regularized EEGNet (KREEGNet), which is a deep learning algorithm for analyzing EEG functional connectivity. The authors evaluated the performance of KREEGNET by making prediction models for MI classification, using two public EEG datasets obtained by motor imaginary tasks. The results showed a good performance as compared with other methods.

This is an interesting paper as an example of improving deep learning methods. Here are my comments;

1. Lines 317-332 explain how to classify subjects into 3 groups. It might be better to move this part to the Materials and Methods section.

2. It might be informative to show the numbers of GI/GII/GIII, although we can count them on Fig. 4.

3. In Lines 270-271, the authors mentioned 'Model performance was evaluated using accuracy, Cohen’s kappa, and the area under the curve ROC'. However, the accuracy is mainly described to explain the model performance, and the latter two seem only used for GI/II/III classification and appear in Table 2. It might be of interest to briefly describe the overall results, including DBI.

4. The hyperparameter settings and Kolmogorov-Smirnov statistic are never mentioned in the Method section. Although it is understandable to report them in the whole context of the study, some readers might feel odd to see they appear suddenly.
